# Circulating Levels of Branched-Chain Amino Acids Are Associated with Diet: A Cross-Sectional Analysis

**DOI:** 10.3390/nu17091471

**Published:** 2025-04-27

**Authors:** Keyuan Liu, Rebecca Borreggine, Hector Gallart-Ayala, Julijana Ivanisevic, Pedro Marques-Vidal

**Affiliations:** 1Department of Medicine, Internal Medicine, Lausanne University Hospital (CHUV) and University of Lausanne, 1011 Lausanne, Switzerland; keyuan.liu@unil.ch; 2Metabolomics Platform, Faculty of Biology and Medicine, University of Lausanne, 1011 Lausanne, Switzerland; rebecca.borreggine@unil.ch (R.B.); hector.gallartayala@unil.ch (H.G.-A.); julijanaivanisevic@unil.ch (J.I.)

**Keywords:** branched-chain amino acids, diet, cross-sectional study

## Abstract

**Background:** Higher circulating branched-chain amino acids (BCAAs) are linked to cardiometabolic and neurological diseases. While diet is the primary BCAA source, its impact on circulating levels remains unclear. This study examined the association between dietary intake and circulating BCAA concentrations in a large population-based sample. **Methods:** Data from 2159 participants (58.2% women, mean age 53.4 ± 8.6 years) were analyzed. Dietary intake was assessed using a questionnaire covering 91 individual food items, 9 nutrient categories, and 3 dietary patterns. BCAA concentrations were measured via LC-MS. All analyses were stratified by gender. **Results:** Circulating BCAA levels were higher in men than in women. BCAA levels were negatively associated with vegetables (standardized β = −0.029, *p* = 0.088; −0.051, *p* = 0.003; −0.038, *p* = 0.043 for leucine, isoleucine, and valine, respectively), dairy (−0.037, *p* = 0.029; −0.063, *p* < 0.001; −0.041, *p* = 0.028), and fruit (−0.031, *p* = 0.084; −0.039, *p* = 0.030; −0.041, *p* = 0.034), and a positive trend was observed for meat and meat-derived products, but the associations did not reach statistical significance. Vegetal protein, total carbohydrates, and monosaccharides showed a significant negative association with circulating BCAAs levels. Participants who complied with “dairy ≥ 3/day”, “meat ≤ 5/week”, or “at least three guidelines” had lower circulating BCAA levels. **Conclusions:** Circulating BCAA levels were negatively associated with dairy, vegetables, fruits, plant protein, carbohydrates, non-digestible fiber, calcium, and iron. While circulating BCAA levels were linked to meat consumption and adherence to dietary guidelines, the association was not linear. Differences were observed between men and women, which may be attributed to variations in dietary intake and preferences.

## 1. Background

Branched-chain amino acids (BCAAs), including isoleucine, leucine, and valine, are essential amino acids with protein anabolic properties [1]. Unlike other amino acids, BCAAs are not directly metabolized by the liver [1]. Instead, ingested BCAAs enter the bloodstream, where they are primarily taken up by skeletal muscle and other tissues. Due to their role in muscle metabolism, BCAAs and BCAA-rich foods or supplements are widely used in sports nutrition [2]. Additionally, BCAAs can influence various diseases through multiple pathways, including the activation of the mTOR pathway [3] and competitive transport across the blood–brain barrier [4]. These effects have been linked to conditions such as obesity, diabetes, hypertension, and other cardiometabolic diseases [5] and neurological disorders [6].

Dietary intake is the primary source of BCAAs. For example, in the Japanese diet, the primary contributors to BCAA intake are cereals, potatoes and starches (23–25%), fish and shellfish (21–23%), and meat (14–15%) [7]. In the United States, the major dietary sources of BCAAs are meat (~37%), milk (~12%), and fish (~8%) [8]. Therefore, understanding the relationship between diet and circulating BCAA levels is crucial. However, existing research findings are inconsistent. Rietman et al. reported that approximately 80% of dietary BCAAs enter circulation [9], while studies on BCAA kinetics suggest an absorption rate of less than 40% [10]. Short-term studies have shown that a high-protein diet (22% of total energy intake) can rapidly increase circulating BCAA levels by 25% in young individuals [11]. In contrast, a longitudinal study conducted by Chinese researchers, tracking dietary intake over one year, found a statistically significant correlation between dietary and circulating BCAA levels [12]. This contrast suggests that while short-term dietary changes may lead to noticeable fluctuations in circulating BCAA levels, the long-term impact of diet on circulatory BCAA levels may be more complex. Furthermore, Merz et al. [13] suggested that fasting levels of circulatory BCAA were associated with the dietary pattern “high in meat, sausages, sauces, eggs, and ice cream, but low in nuts, cereals, mushrooms, and pulses”, rather than only “total protein intake”.

The purpose of this study was to assess the association between dietary intake and circulating BCAA concentrations. The hypothesis was that selected foods would be associated with BCAA levels. To address these purposes, this study collected dietary intake data over an extended period (one month) from a cohort of generally healthy individuals within a specific age range, stratified by sex, and analyzed the relationship between diet and circulating BCAA levels from three perspectives: individual food items, nutrient intake, and overall dietary patterns. By examining each level separately and in combination, we aimed to provide a more comprehensive understanding of how dietary factors influence circulating BCAA levels.

## 2. Methods

### 2.1. Study Design and Participants

The CoLaus|PsyCoLaus study is a population-based study investigating the epidemiology and genetic determinants of psychiatric and cardiovascular disease in Lausanne, Switzerland [14]. Briefly, a representative sample was collected through a simple, non-stratified random sampling of 19,830 individuals (35% of the source population) aged between 35 and 75. The baseline study was conducted between June 2003 and May 2006; the first follow-up was performed between April 2009 and September 2012.

In each survey, participants attended the survey unit in the morning after an 8 h fast. Participants answered questionnaires, underwent a clinical examination, and had their blood collected. Blood samples were drawn for analyses and additional aliquots were stored at −80 °C for BCAA analysis. For this study, data from the first follow-up were considered.

### 2.2. Dietary Intake, Dietary Scores, and Compliance with Dietary Recommendations

Dietary intake was assessed using a self-administered, semi-quantitative food-frequency questionnaire (FFQ), which also included portion size [15]. This FFQ has been validated in the Geneva population [15,16]. Briefly, this FFQ assesses the dietary intake of the previous 4 weeks and consists of 97 different food items that account for more than 90% of the intake of calories, proteins, fat, carbohydrates (including monosaccharides such as sugar and fructose, and polysaccharides such as starch), alcohol, cholesterol, vitamin D, and retinol, and 85% of fiber, carotene, and iron. For each item, consumption frequencies ranging from “less than once during the last 4 weeks” to “2 or more times per day” were provided, and the participants indicated the average serving size (smaller, equal, or bigger) compared to a reference size. Each participant brought along her/his filled-in FFQ, which was checked for completion by trained interviewers the day of the visit. No information regarding which foods were consumed together was available.

For this analysis, all 91 items available on the FFQ, together with 16 macronutrients (including total protein, vegetal protein, animal protein, total carbohydrates, monosaccharides, polysaccharides, total fat, saturated fat, monounsaturated fat, polyunsaturated fat, total fiber, and alcohol) and some micronutrients (including cholesterol, calcium, and iron) extracted from it were used. The different food items were also grouped into categories (i.e., dairy, meat, processed meat, fish, vegetables, and fruits).

Four dietary scores were computed: two based on the alternate healthy eating index (AHEI) and two based on the Mediterranean diet. The first version of the AHEI was adapted from McCullough et al. [17]. In this study, the amount of trans fat could not be assessed, and all participants taking multivitamins were considered as taking them for a duration ≥ 5 years. Thus, the modified AHEI score ranged between 2.5 and 77.5, instead of 2.5 and 87.5 for the original AHEI score. A second version was computed by slightly modifying the scoring of alcohol consumption, using a smooth transition in scoring instead of a yes/no scoring system. The first Mediterranean dietary score (designated as “Mediterranean score 1”) was derived from Trichopoulou et al. [18]. The score uses consumption frequencies instead of amounts. Briefly, a value of 0 or 1 is assigned to each of seven foods using their gender-specific medians as a cut-off. Participants whose consumption frequencies for “healthy” foods (vegetables, fruits, fish, and cereal) were above the median were given the value of 1, while for “unhealthy” foods (meat, dairy products), consumption frequencies below the median were given the value of 1. Two other items were considered: ratio of monounsaturated to saturated fats and moderate alcohol consumption (between 5 and 25 g/day for women and 10 and 50 g/day for men). The score is computed by adding the values for each item and ranges between 0 and 8. The second Mediterranean dietary score (designated as ”Mediterranean score 2”) adapted to the Swiss population was computed according to Vormund et al. [19]. It used the same scoring system but considered nine types of “healthy” foods: fruits, vegetables, fish, cereal, salads, poultry, dairy products, and wine. The score is computed by adding the values for each item and ranges between 0 and 9. For both AHEI and Mediterranean scores, higher values represented a healthier diet.

Participants were also dichotomized according to whether they followed the dietary recommendations for fruits, vegetables, meat, fish, and dairy products from the Swiss Society of Nutrition [20,21]. Compliance was estimated from the FFQ data. The recommendations were ≥2 fruit portions/day; ≥3 vegetable portions/day; ≤5 meat portions/week; ≥1 fish portion/week; and ≥3 dairy products portions/day. In this study, portion size was not used to compute compliance, but relied on consumption frequencies, as there is no perfect match between portion size of the FFQ and portion size of the guidelines, and participants could indicate “smaller” or “bigger” portions without indicating their size. A similar approach was used in other studies [22,23]. This was applied as the portion sizes recommended by the Swiss Society of Nutrition do not take into account a subject’s corpulence and caloric needs [21]. As the FFQ queried about fresh and fried fish, two categories of compliance with fish consumption were considered: one included and one excluded fried fish. For each participant, the number of guidelines complied with was computed. Two sums were computed: one used compliance with fish consumption using all types of fish preparation (i.e., including fried fish); the other used compliance with fish consumption using fresh fish only.

### 2.3. Sample Preparation and Metabolic Profiling

Blood samples were obtained in the morning after an overnight fast using S-Monovette ISO gel tubes (Sarstedt, Nümbrecht, Germany) and allowed to clot. Absolute quantification of BCAA was performed by extracting 20 µL of fasting serum samples with 250 µL of ice-cold methanol, adding the internal standard solution of the corresponding BCAA, and diluting it to 300 µL with 0.1% formic acid in water. After the samples were mixed evenly by shaking, they were centrifuged at 4 °C and 2700× *g* for 15 min. The resulting supernatant was transferred to LC-MS vials prior to injection. A detailed description of the method used has been published [24]. Briefly, a hydrophilic interaction chromatography (HILIC)-based high-resolution mass spectrometry (HRMS) technique was developed, taking advantage of HRMS data acquired in full-scan mode. Accuracy was within 90–106% of validated NIST reference plasma concentrations for the panel of measured BCAA. BCAA extraction recoveries were 87–100% on average, depending on the concentration range spiked. The coefficient of variation (CV) was 1–10% and 1–25% for intra- and interday measurements, respectively.

### 2.4. Other Covariates

Smoking status was self-reported and categorized as never, former, and current. Marital status was categorized as living alone (i.e., single, divorced, or widowed) or living with a partner. Education was categorized into low (compulsory + apprenticeship), medium (high school), or high (university level). Nationality was dichotomized into born in Switzerland or elsewhere. Presence of any type of diet (i.e., for slimming, diabetes, or hypertension) was also collected and categorized as present/absent.

Physical activity was assessed by a questionnaire validated in the population of Geneva [25]. This self-reported questionnaire assesses the type and duration of 70 kinds of professional and non-professional activities and sports during the previous week. Sedentary status was defined as spending over 90% of the daily energy in activities below moderate- and high-intensity (defined as requiring at least 4 times the basal metabolic rate [26]).

Body weight and height were measured with participants barefoot and in light indoor clothes. Body weight was measured in kilograms to the nearest 100 g using a Seca^®^ scale (Hamburg, Germany). Height was measured to the nearest 5 mm using a Seca^®^ (Hamburg, Germany) height gauge. Body mass index (BMI) was computed and categorized into normal (BMI < 25 kg/m^2^), overweight (25 ≤ BMI < 30 kg/m^2^), and obese (BMI ≥ 30 kg/m^2^).

### 2.5. Eligibility and Exclusion Criteria

Participants with BCAA levels were considered as eligible. Participants were excluded if they have no dietary markers, if they presented extreme total energy intake values (defined as <500 or >3500 kcal/day for women and <800 or >4000 kcal/day for men), if they missed any confounder variable, or if they reported being on any type of diet (for slimming, antidiabetic, etc.).

### 2.6. Statistical Analyses

Statistical analysis was conducted using Stata version 17.0 (Stata Corp, College Station, TX, USA). Summary statistics are reported as mean ± standard deviation for continuous variables and as number of participants (percentage) for categorical variables. Bivariate between-group comparisons were performed using *t*-test for continuous variables and chi-square for categorical variables. Normality of the distribution of BCAA was checked via Q-Q plots, and no deviation was found.

As dietary intake values presented with a skewed distribution, bivariate associations between dietary markers and the levels of circulatory BCAA were computed using Spearman correlations for the overall sample and stratified by gender. Multivariable regression analysis assessing the association between dietary markers and levels of circulatory BCAA were conducted using linear regression adjusting on age (continuous), smoking status (never, former, current), educational level (low, medium, high), marital status (yes, no), born in Switzerland (yes, no), presence of a diet (yes, no), BMI categories (normal, overweight), and sedentary status (yes, no). Results were presented as standardized beta coefficients, as their interpretation is similar to the correlation coefficients.

Bivariate analysis of the associations between dietary recommendations and levels of circulatory BCAA were assessed by comparing levels of circulatory BCAA between compliant and non-compliant participants using Student’s *t*-test, and the results were expressed as mean ± standard deviation; multivariable analysis was conducted using analysis of variance, adjusting for the same variables as above, and results were expressed as multivariable-adjusted mean ± standard error. As this was an exploratory study and similar to other studies [27,28], we decided not to correct for multiple testing, and we defined statistical significance as a two-sided test with *p* < 0.05.

## 3. Results

### 3.1. Selection of Participants

Of the initial 5064 participants, 2577 (50.9%) were eligible for BCAA analysis. Of them, 2159 (83.8%) were included in this study. The reasons for exclusion are indicated in Appendix A and the characteristics of the included and the excluded participants are indicated in Appendix A. Included participants were more frequently women (1257 [58.2] vs. 1450 [49.9]%, *p* < 0.001), younger (53.4 ± 8.6 vs. 61.0 ± 10.7 years, *p* < 0.001), with a higher educational level (600 [27.8] vs. 479 [16.5]%, *p* < 0.001), less frequently former smokers (748 [34.7] vs. 1135 [39.9]%, *p* < 0.001), on a diet (476 [22.1] vs. 1098 [37.8]%, *p* < 0.001) or sedentary (1082 [50.1] vs. 1323 [65.4]%, *p* < 0.001), and none had obesity.

Table 1 presents the characteristics of the sample according to gender. Women were older (54.3 ± 8.8 vs. 52.2 ± 8.2 years, *p* < 0.001), less frequently living in a couple (643 [51.2] vs. 604 [67.0]%, *p* < 0.001), less frequently of higher education (300 [23.9] vs. 300 [33.3]%, *p* < 0.001), more frequently on a diet (316 [25.1] vs. 160 [17.7]%, *p* < 0.001) or sedentary (704 [56.0] vs. 378 [41.9]%, *p* < 0.001), and less frequently overweight (363 [28.9] vs. 450 [49.9]%, *p* < 0.001). BCAA levels were higher in men than in women: valine (250.2 ± 45.2 vs. 207.9 ± 36.1 µmol/L, *p* < 0.001); leucine (136.0 ± 24.4 vs. 106.3 ± 17.4 µmol/L, *p* < 0.001); and isoleucine (63.6 ± 13.0 vs. 48.1 ± 9.2 µmol/L, *p* < 0.001).

### 3.2. Associations Between Individual Food Items, Food Categories, and BCAA

The bivariate associations between intake of 91 individual food items and levels of circulatory BCAA are shown in Appendix A for all the individual values of the Spearman correlations. Dairy products (yogurt and cheese), bread and cereals, fruits and vegetables were negatively associated with BCAA levels, while meat, sugary products (honey and jam), and alcoholic beverages (wine, beer, and spirits) were positively associated with BCAA, the strongest association being observed for vegetables, meat, and alcoholic beverages. Overall, the associations between foods and BCAA levels tended to be stronger in men than in women (Appendix A).

After multivariable analysis, except for the negative association between fruit, vegetable, water intake, and levels of circulatory BCAA, most associations were no longer statistically significant, which was not present before (Appendix A).

The bivariate associations between food categories and BCAA are shown in Appendix A. Positive associations were found for meat and processed meat, and negative associations with vegetables and fruits, and similar albeit non-significant associations were found when the analysis was split according to gender. The results of the multivariable associations between food categories and BCAA are summarized in Table 2. Negative associations were found for dairy, vegetables, and fruits, but no longer for meat or processed meat. Dairy was negatively associated with leucine, isoleucine, and valine; vegetables were negatively associated with isoleucine and valine; and fruits were also negatively associated with isoleucine and valine. Similar results were obtained when the analysis was split according to gender.

### 3.3. Associations Between Nutrients and BCAA

The bivariate associations between 15 nutrients and levels of circulatory BCAA are shown in Appendix A. Positive associations were found between BCAA and most nutrients, excepting total fiber (negative association), monosaccharides, and calcium (no association).

After multivariable analysis, the results of the associations between nutrients and levels of circulatory BCAA are summarized in Table 3. Negative associations were found between all three BCAA and vegetal protein, total carbohydrates, monosaccharides and polysaccharides, and total fiber, and similar results were obtained when the analysis was split according to gender.

### 3.4. Associations Between Dietary Scores and BCAA

The bivariate relationship between four dietary scores and levels of circulatory BCAA is shown in Appendix A. Both AHEIs were negatively associated with all BCAA, while no association was found for the Mediterranean diet scores. When the analysis was split by gender, similar associations were observed, although they did not reach statistical significance.

After multivariable analysis, the results are summarized in Table 4. Again, significant negative associations were found between both versions of AHEI and isoleucine and valine in the whole sample, while no associations were found for the Mediterranean diet scores or for leucine. When the analysis was split by gender, similar associations were observed, although they did not reach statistical significance.

### 3.5. Associations Between Compliance with Dietary Recommendations and BCAA

The bivariate associations between compliance with dietary recommendations and BCAA are summarized in Appendix A. Compliance with the dietary recommendations led to lower BCAA levels, namely for isoleucine, and this association was also present in men but not in women.

After multivariable analysis, the results are summarized in Table 5. Participants who complied with dairy and meat recommendations and at least three recommendations (at least three guidelines with all types of fish; at least three guidelines excluding fried fish) had lower BCAA levels; similar significant results were obtained in men and similar albeit non-significant results were obtained in women.

## 4. Discussion

To the best of our knowledge, this is one of the largest studies to evaluate the correlation between circulating BCAA levels and dietary intake in the general population. We examined this association from individual food and nutrient intakes, and dietary patterns from three different perspectives; further analysis was conducted separately by gender.

### 4.1. Associations Between Individual Food Items, Food Categories, and BCAA

Levels of circulatory BCAAs are highly dependent on protein source [8], but whether levels of circulatory BCAA individually reflect protein intake have been insufficiently studied. In this study, we found that dairy products (yogurt and cheese), bread and cereals, and fruits and vegetables were negatively associated with BCAA levels, while meat, sugary products (honey and jam), and alcoholic beverages (wine, beer, and spirits) showed a positive but inconsistent association with levels of circulatory BCAA. The negative association with fruits and vegetables agrees with the low BCAA content of these food groups. In addition, we also found associations between different beverage consumption and levels of circulatory BCAA, such as tea showing a negative association, and wine showing a positive association.

Interestingly, in this study, the association between meat and processed meat consumption and circulatory BCAA levels was relatively weak. This may be related to the selection criteria of the study population, as only apparently healthy participants were eligible for circulatory BCAA assessment. Indeed, Rousseau et al. [29] reported that the positive association between animal protein or red meat intake and levels of circulatory BCAA was found in overweight people or individuals with metabolic syndrome, but not in healthy individuals with normal weight. They believe that gender seems to be a key factor in this association. In healthy individuals, BCAA may be more influenced by gender (and other metabolic factors) than by protein source intake. Similar results have also been reported by Schmidt et al., who conducted a cross-sectional analysis on the relationship between plasma amino acid concentrations in 392 participants with different dietary habits, targeting the male population. The results also showed that there was no significant difference in circulatory BCAA concentrations among meat eaters, fish, vegetarians, and vegans, and even the circulatory BCAA concentrations of fish eaters and vegetarians were slightly higher than those of meat eaters [30].

We also conducted separate research and analysis based on gender differences. Compared with female participants, the levels of circulatory BCAA in male participants was more closely related to diet. Among male participants, meat and derived products all showed a significant positive correlation with BCAA, but these were not found in female participants. The negative correlation between kiwi and BCAA in female participants was not found in male participants. After multivariable analysis, the trend associated with circulatory BCAA concentrations disappeared in male participants, but in female participants, fruit/aroma yogurt and berries (strawberries, blueberries) still showed a significant negative association with the levels of circulatory BCAA. Therefore, in the female population, those who have a dietary habit of eating more yogurt and fruits may have lower levels of circulatory BCAA.

These performances may be related to the different dietary preferences of men and women; other possible explanations include the higher BMI among males and the higher prevalence of diets among females. Since the difference between men and women disappeared after grouping the foods, we believe the initial disparity may have been due to the small intake of individual foods, leading to less reliable results. By grouping similar foods into broader categories, the total intake within each category increased, improving the statistical robustness and the accuracy of the results.

### 4.2. Associations Between Nutrients and BCAA

Strong negative associations were found between circulatory BCAA levels and vegetal protein, total carbohydrates, monosaccharides, polysaccharides, total non-digestible fiber, calcium, and iron, while no association was observed with animal protein intake. Our findings are somewhat different from those of Rousseau et al. [29] and Hamaya et al. [31], who reported positive associations between levels of circulatory BCAA and animal protein but not with plant protein. This discrepancy may be attributed to differences in the study populations; participants with higher animal protein intake may be more prone to obesity, while our study focused on a relatively healthy population, excluding obese individuals. Additionally, the sources of plant protein could vary; for instance, mung beans provide not only plant protein but also significant amounts of dietary fiber, which may influence the results. Moreover, different animal- or plant-based diets consistently change the composition of the gut microbiota, which is also a source of variable circulatory BCAA levels; those changes could also influence the associations between nutrients and levels of circulatory BCAA [32,33,34].

It is worth noting that although the trends are similar, the association between iron and BCAA in women and the association between polysaccharides and BCAA in men did not reach statistical significance. Possible reasons for these differences include, on the one hand, the influence of women’s physiological cycles and sex hormones, which increase their iron requirements while often resulting in insufficient intake, thereby affecting the outcomes. On the other hand, differences in dietary preferences between men and women may also play a role, as men tend to consume lower amounts of polysaccharides, which could influence the results.

### 4.3. Associations Between Dietary Scores and BCAA

Our results showed a significant negative association between AHAI scores and levels of circulatory BCAA in the entire sample. However, this association was no longer observed when the analysis was stratified by gender, and no association was found with Mediterranean diet scores.

Hamaya et al. [31], who also analyzed the association between the 2010 Alternative Healthy Diet Index (non-alcoholic) and the Alternative Mediterranean Diet Score with BCAA levels in women, similarly found no significant association between these dietary scores and BCAA levels.

This is different from the findings of Canela et al. [35], who conducted a dietary intervention on 251 type 2 diabetes patients and 694 controls and found that the Mediterranean diet rich in extra virgin olive oil significantly reduced the plasma BCAA levels. Possible explanations for our results include the analysis of a free-living, non-intervention sample, or that we only counted the participants’ diet in the past four weeks instead of a whole year, as in the study by Canela et al.

### 4.4. Associations Between Compliance with Dietary Recommendations and BCAA

The results show that those men participants who were compliers with the guidelines of “meat ≤ 5/week”, “dairy ≥ 3/day”, or “at least 3 guidelines” had lower circulating BCAA levels compared to non-compliers. Interestingly, when we analyzed meat consumption as a continuous variable at the levels of individual foods and nutrient intakes, no significant association with BCAA levels was found. Similarly, no association was observed when we analyzed dietary patterns (Mediterranean diet and AHEI scores) as continuous variables. Therefore, we conclude that while meat consumption and dietary patterns are significantly associated with BCAA levels, these associations do not appear to be linear.

In addition, no significant association was observed in females, possibly due to their relatively smaller meal intakes. This may have led to smaller differences between the compliance and non-compliance groups, thereby influencing the results.

Combined with the previous findings, BCAA levels were negatively associated with dairy, vegetables, fruits, plant protein, carbohydrates, non-digestible fiber, calcium, and iron. While BCAA levels were linked to meat consumption and adherence to dietary guidelines, the relationship was not linear. Differences were observed between men and women, which may be attributed to variations in dietary intake and preferences.

### 4.5. Strengths and Limitations

This study has several strengths. First, it included only healthy, community-dwelling participants, thus allowing for establishment of the associations between levels of circulatory BCAA and dietary intake in a group devoid of major cardiovascular risk factors. Second, it is the most comprehensive regarding food items, and one of the largest regarding sample size (N = 2159), being outnumbered only by the studies of Floegel et al. (N = 2380) [36] and Hamaya et al. (N = 18,897) [31], although the latter study only included women.

This study also has some limitations. First, it was conducted in a specific region, and we know that diet changes according to region in Switzerland [37]. Hence, the results might differ depending on the geographical region, although our results agree with those of studies conducted in other countries. Second, participants were selected and apparently healthy (i.e., not diabetic, not obese). Still, our findings agree with other studies that included people with diabetes or obesity [29]. We did not correct for multiple testing, which would reduce the number of significant associations. Still, this was also the approach conducted by other studies [28]. Dietary intake was based on self-reported data, which might not fully correspond to actual dietary intake. Still, it would be impractical and very time-consuming to assess dietary intake in a large study using other methods such as 24 h recalls. Finally, the differences in levels of circulatory BCAA observed were small, usually less than 5 μmol/L, and might not be clinically relevant. Still, they suggest that dietary patterns might influence levels of circulatory BCAA.

## 5. Conclusions

BCAA levels were negatively associated with dairy, vegetables, fruits, plant protein, carbohydrates, non-digestible fiber, calcium, and iron. While BCAA levels were linked to meat consumption and adherence to dietary guidelines, the relationship was not linear. Differences were observed between men and women, which may be attributed to variations in dietary intake and preferences.

## Figures and Tables

**Table 1 nutrients-17-01471-t001:** Characteristics of the sample, overall and by gender.

	Women	Men	*p*-Value
Sample size	1257	902	
Age (years)	54.3 ± 8.8	52.2 ± 8.2	<0.001
Born in Switzerland (%)	814 (64.8)	557 (61.8)	0.153
Living in a couple (%)	643 (51.2)	604 (67.0)	<0.001
Educational level (%)			<0.001
High	300 (23.9)	300 (33.3)	
Medium	376 (29.9)	236 (26.2)	
Low	581 (46.2)	365 (40.5)	
Smoking categories (%)			0.833
Never	541 (43.0)	377 (41.8)	
Former	430 (34.2)	318 (35.3)	
Current	286 (22.8)	207 (23.0)	
On a diet § (%)	316 (25.1)	160 (17.7)	<0.001
Sedentary (%)	704 (56.0)	378 (41.9)	<0.001
Body mass index (kg/m^2^)	23.3 ± 3.0	24.9 ± 2.5	<0.001
BMI categories (%)			<0.001
Normal	894 (71.1)	452 (50.1)	
Overweight	363 (28.9)	450 (49.9)	
BCAA (µmol/L)			
Valine	207.9 ± 36.1	250.2 ± 45.2	<0.001
Leucine	106.3 ± 17.4	136.0 ± 24.4	<0.001
Isoleucine	48.1 ± 9.2	63.6 ± 13.0	<0.001

§, low calorie, low salt, for diabetes or dyslipidaemia. BMI, body mass index; BCAA, branched-chain amino acids. Results are expressed as number of participants (column percentage) for categorical variables and as average ± standard deviation for continuous variables. Between-group comparisons performed using chi-square for categorical variables and Student’s *t*-test for continuous variables.

**Table 2 nutrients-17-01471-t002:** Multivariable analysis of the associations between blood levels of branched-chain amino acids and the main food groups, CoLaus|PsyColaus study, Lausanne, Switzerland, 2009–2012, overall and stratified by gender.

	Leucine		Isoleucine		Valine	
Food Groups, gr/day	Beta	*p*-Value	Beta	*p*-Value	Beta	*p*-Value
All						
Dairy	−0.037	0.029	−0.063	<0.001	−0.041	0.028
Meat	0.026	0.140	0.018	0.303	0.032	0.096
Processed meat	0.022	0.200	0.022	0.213	0.028	0.143
Fish	0.000	0.978	−0.008	0.625	0.013	0.501
Vegetables	−0.029	0.088	−0.051	0.003	−0.038	0.043
Fruits	−0.031	0.084	−0.039	0.030	−0.041	0.034
Women						
Dairy	−0.074	0.019	−0.065	0.019	−0.082	0.010
Meat	0.032	0.312	0.019	0.501	0.031	0.340
Processed meat	0.026	0.415	0.012	0.675	0.042	0.189
Fish	0.032	0.321	−0.028	0.320	0.040	0.207
Vegetables	−0.032	0.323	−0.064	0.022	−0.035	0.277
Fruits	−0.022	0.496	−0.076	0.007	−0.023	0.484
Men						
Dairy	−0.023	0.419	−0.095	0.003	−0.020	0.479
Meat	0.026	0.348	0.022	0.499	0.045	0.107
Processed meat	0.024	0.384	0.033	0.307	0.012	0.665
Fish	−0.038	0.177	0.003	0.934	−0.017	0.555
Vegetables	−0.039	0.163	−0.063	0.050	−0.047	0.091
Fruits	−0.053	0.063	−0.015	0.643	−0.064	0.024

Results are expressed as standardized beta coefficients. Analysis conducted using linear regression and adjusting for gender (ALL only), age (continuous), smoking (never, former, current), educational level (high, medium, low), marital status (in couple, alone), born in Switzerland (yes, no), on a diet (yes, no), BMI categories (normal, overweight), and sedentary status (yes, no). Statistically significant coefficients (*p* < 0.05) are indicated in red.

**Table 3 nutrients-17-01471-t003:** Multivariable analysis of the associations between blood levels of branched-chain amino acids and nutrients, CoLaus|PsyColaus study, Lausanne, Switzerland, 2009–2012, overall and stratified by gender.

	Leucine		Isoleucine		Valine	
Nutrients	Beta	*p*-Value	Beta	*p*-Value	Beta	*p*-Value
All						
Total protein, gr/day	−0.015	0.405	−0.041	0.023	−0.007	0.717
Vegetal protein, gr/day	−0.053	0.003	−0.068	<0.001	−0.041	0.001
Animal protein, gr/day	0.004	0.841	−0.021	0.238	0.017	0.384
Total carbohydrates, gr/day	−0.063	<0.001	−0.041	<0.001	−0.041	<0.001
Monosaccharides, gr/day	−0.057	0.001	−0.072	<0.001	−0.041	<0.001
Polysaccharides, gr/day	−0.045	0.012	−0.056	0.002	−0.051	0.010
Total fat, gr/day	−0.019	0.274	−0.030	0.087	−0.024	0.211
Saturated fat (SFA), gr/day	−0.027	0.135	−0.042	0.020	−0.031	0.109
Monounsaturated fat (MUFA), gr/day	−0.010	0.563	−0.018	0.293	−0.012	0.535
Polyunsaturated fat (PUFA), gr/day	−0.011	0.545	−0.011	0.539	−0.025	0.200
Total non-digestible fiber, gr/day	−0.042	* 0.016 *	−0.060	* 0.001 *	−0.055	* 0.004 *
Cholesterol, mg/day	−0.004	*0.822*	−0.018	*0.300*	0.004	*0.839*
Alcohol, mL/day	0.027	*0.132*	0.027	*0.136*	−0.014	*0.492*
Calcium, mg/day	−0.031	*0.068*	−0.058	* 0.001 *	−0.024	*0.199*
Iron, mg/day	−0.017	*0.319*	−0.040	* 0.020 *	−0.032	*0.093*
Women						
Total protein, gr/day	−0.019	*0.549*	−0.055	*0.084*	−0.012	*0.700*
Vegetal protein, gr/day	−0.071	* 0.025 *	−0.097	* 0.002 *	−0.065	* 0.042 *
Animal protein, gr/day	0.004	*0.898*	−0.030	*0.355*	0.010	*0.757*
Total carbohydrates, gr/day	−0.085	* 0.008 *	−0.104	* 0.001 *	−0.088	* 0.006 *
Monosaccharides, gr/day	−0.084	* 0.008 *	−0.086	* 0.007 *	−0.096	* 0.003 *
Polysaccharides, gr/day	−0.060	*0.059*	−0.090	* 0.005 *	−0.053	*0.095*
Total fat, gr/day	−0.042	*0.188*	−0.059	*0.066*	−0.048	*0.129*
Saturated fat (SFA), gr/day	−0.044	*0.164*	−0.069	* 0.031 *	−0.051	*0.110*
Monounsaturated fat (MUFA), gr/day	−0.035	*0.266*	−0.045	*0.158*	−0.037	*0.242*
Polyunsaturated fat (PUFA), gr/day	−0.029	*0.357*	−0.037	*0.253*	−0.050	*0.118*
Total non-digestible fiber, gr/day	−0.057	*0.079*	−0.071	* 0.030 *	−0.045	*0.163*
Cholesterol, mg/day	−0.004	*0.902*	−0.031	*0.333*	−0.001	*0.976*
Alcohol, mL/day	0.026	*0.434*	0.026	*0.436*	−0.014	*0.663*
Calcium, mg/day	−0.058	*0.069*	−0.090	* 0.005 *	−0.049	*0.127*
Iron, mg/day	−0.015	*0.630*	−0.043	*0.175*	−0.021	*0.507*
Men						
Total protein, gr/day	−0.021	*0.459*	−0.041	*0.139*	−0.005	*0.858*
Vegetal protein, gr/day	−0.046	*0.099*	−0.055	*0.051*	−0.067	* 0.018 *
Animal protein, gr/day	−0.005	*0.853*	−0.026	*0.351*	0.022	*0.437*
Total carbohydrates, gr/day	−0.058	* 0.040 *	−0.072	* 0.010 *	−0.083	* 0.003 *
Monosaccharides, gr/day	−0.056	* 0.045 *	−0.088	* 0.002 *	−0.083	* 0.003 *
Polysaccharides, gr/day	−0.035	*0.216*	−0.026	*0.357*	−0.048	*0.089*
Total fat, gr/day	0.002	*0.947*	−0.002	*0.930*	−0.001	*0.975*
Saturated fat (SFA), gr/day	−0.012	*0.675*	−0.016	*0.555*	−0.014	*0.630*
Monounsaturated fat (MUFA), gr/day	0.014	*0.613*	0.007	*0.789*	0.012	*0.660*
Polyunsaturated fat (PUFA), gr/day	0.011	*0.687*	0.023	*0.419*	0.003	*0.912*
Total non-digestible fiber, gr/day	−0.049	*0.081*	−0.079	* 0.005 *	−0.077	* 0.006 *
Cholesterol, mg/day	−0.008	*0.781*	−0.013	*0.643*	0.009	*0.736*
Alcohol, mL/day	0.050	*0.075*	0.047	*0.091*	−0.003	*0.904*
Calcium, mg/day	−0.025	*0.372*	−0.056	* 0.043 *	−0.014	*0.621*
Iron, mg/day	−0.030	*0.290*	−0.057	* 0.042 *	−0.050	*0.077*

Results are expressed as standardized beta coefficients. Analysis conducted using linear regression and adjusting for gender (ALL only), age (continuous), smoking (never, former, current), educational level (high, medium, low), marital status (in couple, alone), born in Switzerland (yes, no), on a diet (yes, no), BMI categories (normal, overweight), and sedentary status (yes, no). Statistically significant coefficients (*p* < 0.05) are indicated in red.

**Table 4 nutrients-17-01471-t004:** Multivariable analysis of the associations between blood levels of branched-chain amino acids and dietary scores, CoLaus|PsyColaus study, Lausanne, Switzerland, 2009–2012, overall and stratified by gender.

	Leucine		Isoleucine		Valine	
	Beta	*p*-Value	Beta	*p*-Value	Beta	*p*-Value
All						
Mediterranean diet score (Trichopoulou)	−0.011	0.516	−0.016	0.362	−0.020	0.299
Mediterranean diet score (Vormund)	0.001	0.974	−0.019	0.283	−0.008	0.657
Alternate healthy eating index (Version 1)	−0.025	0.153	−0.042	0.017	−0.040	0.040
Alternate healthy eating index (Version 2)	−0.026	0.146	−0.043	0.016	−0.040	0.041
Women						
Mediterranean diet score (Trichopoulou)	−0.032	0.325	−0.035	0.281	−0.032	0.322
Mediterranean diet score (Vormund)	−0.006	0.862	−0.031	0.334	−0.008	0.797
Alternate healthy eating index (Version 1)	−0.038	0.253	−0.059	0.074	−0.038	0.250
Alternate healthy eating index (Version 2)	−0.038	0.251	−0.058	0.078	−0.038	0.255
Men						
Mediterranean diet score (Trichopoulou)	0.007	0.801	−0.001	0.982	−0.007	0.803
Mediterranean diet score (Vormund)	0.005	0.855	−0.016	0.569	−0.008	0.791
Alternate healthy eating index (Version 1)	−0.020	0.476	−0.043	0.134	−0.044	0.125
Alternate healthy eating index (Version 2)	−0.022	0.433	−0.045	0.117	−0.045	0.117

Results are expressed as standardized beta coefficients. Analysis conducted using linear regression adjusting for gender (ALL only), age (continuous), smoking (never, former, current), educational level (high, medium, low), marital status (in couple, alone), born in Switzerland (yes, no), on a diet (yes, no), BMI categories (normal, overweight), and sedentary status (yes, no). Statistically significant coefficients (*p* < 0.05) are indicated in red.

**Table 5 nutrients-17-01471-t005:** Multivariable analysis of blood levels in μmol/L of branched-chain amino acids according to compliance with Swiss dietary guidelines, CoLaus|PsyColaus study, Lausanne, Switzerland, 2009–2012, overall and stratified by gender.

	Leucine			Isoleucine			Valine		
Gender/Guideline	Non Complier	Complier	*p*-Value	Non Complier	Complier	*p*-Value	Non Complier	Complier	*p*-Value
All									
Fruits ≥ 2/day	119.1 ± 0.6	118.1 ± 0.7	0.281	54.9 ± 0.3	54.3 ± 0.4	0.201	226.5 ± 1.1	224.1 ± 1.3	0.174
Vegetables ≥ 3/day	118.8 ± 0.4	117.2 ± 1.6	0.337	54.8 ± 0.2	53.1 ± 0.9	0.071	225.9 ± 0.9	220.8 ± 3.2	0.129
Meat ≤ 5/week	120.0 ± 0.7	117.9 ± 0.6	0.020	54.9 ± 0.4	54.5 ± 0.3	0.430	227.6 ± 1.4	224.2 ± 1.1	0.056
Fish all ≥ 1/week	118.8 ± 0.7	118.7 ± 0.5	0.917	55.0 ± 0.4	54.5 ± 0.3	0.318	225.0 ± 1.5	225.8 ± 1.0	0.651
Fish not fried ≥ 1/week	118.5 ± 0.6	119.0 ± 0.7	0.592	54.7 ± 0.3	54.5 ± 0.4	0.574	224.9 ± 1.1	226.4 ± 1.3	0.378
Dairy ≥ 3/day	119.0 ± 0.4	115.7 ± 1.5	0.031	54.9 ± 0.2	52.0 ± 0.8	<0.001	226.1 ± 0.9	219.1 ± 2.9	0.019
At least 3 guidelines ^a^	119.2 ± 0.5	117.0 ± 0.9	0.029	55.0 ± 0.3	53.5 ± 0.5	0.005	226.8 ± 1.0	221.5 ± 1.8	0.010
At least 3 guidelines ^b^	119.0 ± 0.5	117.2 ± 1.0	0.122	54.9 ± 0.3	53.4 ± 0.6	0.014	226.3 ± 0.9	221.9 ± 2.0	0.048
Women									
Fruits ≥ 2/day	106.6 ± 0.7	105.8 ± 0.7	0.429	48.5 ± 0.4	47.7 ± 0.4	0.123	209.3 ± 1.4	206.3 ± 1.5	0.148
Vegetables ≥ 3/day	106.2 ± 0.5	105.9 ± 1.6	0.865	48.2 ± 0.3	47.2 ± 0.8	0.276	208.3 ± 1.1	204.1 ± 3.4	0.239
Meat ≤ 5/week	106.8 ± 0.9	105.9 ± 0.6	0.400	48.2 ± 0.5	48.1 ± 0.3	0.829	208.7 ± 1.8	207.6 ± 1.2	0.602
Fish all ≥ 1/week	106.3 ± 0.8	106.2 ± 0.6	0.878	48.4 ± 0.4	47.9 ± 0.3	0.421	208.2 ± 1.7	207.8 ± 1.2	0.840
Fish not fried ≥ 1/week	106.1 ± 0.7	106.3 ± 0.7	0.826	48.2 ± 0.3	48.0 ± 0.4	0.714	207.7 ± 1.4	208.2 ± 1.5	0.778
Dairy ≥ 3/day	106.2 ± 0.5	106.2 ± 1.6	0.975	48.2 ± 0.3	46.6 ± 0.9	0.061	207.9 ± 1.1	208.2 ± 3.4	0.925
At least 3 guidelines ^a^	106.4 ± 0.6	105.7 ± 0.9	0.493	48.4 ± 0.3	47.4 ± 0.5	0.072	208.7 ± 1.2	206.0 ± 1.9	0.226
At least 3 guidelines ^b^	106.3 ± 0.5	105.8 ± 1.0	0.693	48.3 ± 0.3	47.3 ± 0.5	0.097	208.4 ± 1.1	206.3 ± 2.2	0.407
Men									
Fruits ≥ 2/day	136.0 ± 0.9	135.1 ± 1.4	0.585	63.6 ± 0.5	63.5 ± 0.7	0.900	249.9 ± 1.7	249.0 ± 2.5	0.775
Vegetables ≥ 3/day	136.0 ± 0.8	130.4 ± 3.9	0.159	63.7 ± 0.4	60.0 ± 2.1	0.074	249.9 ± 1.5	242.8 ± 7.2	0.332
Meat ≤ 5/week	137.4 ± 1.1	134.1 ± 1.1	0.033	63.9 ± 0.6	63.3 ± 0.6	0.470	252.6 ± 2.0	246.7 ± 2.0	0.044
Fish all ≥ 1/week	135.8 ± 1.3	135.7 ± 0.9	0.955	63.9 ± 0.7	63.4 ± 0.5	0.554	248.1 ± 2.5	250.3 ± 1.7	0.470
Fish not fried ≥ 1/week	135.5 ± 1.0	136.1 ± 1.3	0.721	63.7 ± 0.5	63.3 ± 0.7	0.613	248.8 ± 1.8	251.0 ± 2.4	0.450
Dairy ≥ 3/day	136.4 ± 0.8	128.2 ± 2.7	0.004	64.0 ± 0.4	59.2 ± 1.4	0.002	251.1 ± 1.5	232.5 ± 5.0	<0.001
At least 3 guidelines ^a^	136.6 ± 0.8	131.6 ± 1.9	0.018	64.0 ± 0.4	61.6 ± 1.0	0.038	251.3 ± 1.6	241.3 ± 3.5	0.011
At least 3 guidelines ^b^	136.3 ± 0.8	131.7 ± 2.3	0.067	63.9 ± 0.4	61.4 ± 1.2	0.068	250.7 ± 1.5	240.8 ± 4.3	0.031

^a^, using all types of fish; ^b^, excluding fried fish. Results are expressed as multivariable adjusted mean ± standard error. Between-group comparisons performed using analysis of variance adjusting for gender (ALL only), age (continuous), smoking (never, former, current), educational level (high, medium, low), marital status (in couple, alone), born in Switzerland (yes, no), on a diet (yes, no), BMI categories (normal, overweight), and sedentary status (yes, no). Statistically significant (*p* < 0.05) are indicated in red.

## Data Availability

The data of CoLaus|PsyCoLaus study used in this article cannot be fully shared as they contain potentially sensitive personal information on participants. According to the Ethics Committee for Research of the Canton of Vaud, sharing these data would be a violation of the Swiss legislation with respect to privacy protection. However, coded individual-level data that do not allow researchers to identify participants are available upon request to researchers who meet the criteria for data sharing of the CoLaus|PsyCoLaus Datacenter (CHUV, Lausanne, Switzerland). Any researcher affiliated to a public or private research institution who complies with the CoLaus|PsyCoLaus standards can submit a research application to research.colaus@chuv.ch or research.psycolaus@chuv.ch. Proposals requiring baseline data only will be evaluated by the baseline (local) Scientific Committee (SC) of the CoLaus and PsyCoLaus studies. Proposals requiring follow-up data will be evaluated by the follow-up (multicentric) SC of the CoLaus|PsyCoLaus cohort study. Detailed instructions for gaining access to the CoLaus|PsyCoLaus data used in this study are available at www.colaus-psycolaus.ch/professionals/how-to-collaborate/ (accessed on 12 June 2024).

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
