# Peer review of "Circulating Levels of Branched-Chain Amino Acids Are Associated with Diet: A Cross-Sectional Analysis"

_nutrients, 2025, doi:10.3390/nu17091471_

Round 1

Reviewer 1 Report

Comments and Suggestions for Authors

Hello

Read your paper with interest.  I have both general and specific comments

Specific:

Diet based studies are challenging.  I understand looking at high meat vs. "lower meat" diets and looking for correlations with BCAA levels.  I don't understand looking for correlations at the level of dairy, vegetables, fruits etc.

What wasn't clear to me was a few things: 1) how can your parse out correlations to one food type (e.g., fruits or vegetable).  People usually don't eat just one food.  2) Why would we care about looking at correlations at foods that are known to have low to no BCAA in them?

I think doing the correlations for the dietary patterns is worthwhile, I am not sure that you can have sound data looking at the individual food group level

Title: diet (not dietary)

Fix references from superscript to normal size

Avoid 1st person language (get rid of "we")

Intro paragraph - it would be good if it read "The purpose of this study......"  And it would be valuable to have hypotheses presented

L55: ...sex. And analysed (why the capitalized And).  Also and should not start a sentence

L86: its standard, when using i.e., to have that in ( ).  (i.e., dairy, meat, ...

Is consumption frequency an issue?  Seems like if you focused on size that would help you to estimate the amount of BCAA consumption better than frequency.  Is this a validated approach?  Any citations to support?

Change multivariable to multivariate

L171: lowercase multivariable

Capitalize table title first word

L210: lowercase multi

L288: correlation better word than association

L351: what is the role of the "n"

Reviewer 2 Report

Comments and Suggestions for Authors

Circulating levels of branched-chain amino acids are associated with dietary: A cross-sectional analysis

Higher circulating branched-chain amino acids (BCAAs) are linked to cardiometabolic and neurological diseases. This study examined the relationship between dietary intake and circulating concentrations of branched-chain amino acids (BCAAs) in a large population-based sample. The authors concluded that circulating BCAA levels were negatively associated with the consumption of dairy products, vegetables, fruits, plant-based protein, carbohydrates, fiber, calcium, and iron. Differences were observed between men and women, which may be attributed to variations in dietary intake and preferences. 

The abstract lacks statistical statements and P values for significant terms.

L46: found only a weak but statistically significant correlation between dietary and circulating BCAA levels (10). Revise the statement. Since it was significantly different, it's not weak. 

In the introduction, you need to provide a paragraph to compare BCAAs from plant-based vs. meat sources with actual numbers.

L84-85: provide the macro and micronutrients that were tested

Could you provide more details about the scoring system, for example, for Mediterranean scores 1 and 2 

L116: Provide details about blood samples and the process to get serum.

L147: revise, change will

L178-179: revise

L337: monosaccharides, polysaccharides, like what? how they are different from carbohydrates.

L412: Fiber is part of carbohydrates; you need to be specific
